# VCR-Graphormer: A Mini-batch Graph Transformer via Virtual Connections

**Dongqi Fu**[*], **Zhigang Hua**[†], **Yan Xie**[†], **Jin Fang**[†], **Si Zhang**[†], **Kaan Sancak**[‡], **Hao Wu**[†],
**Andrey Malevich**[†], **Jingrui He**[*], **Bo Long**[†]
[*]University of Illinois Urbana-Champaign, [†]Meta AI, [‡]Georgia Institute of Technology
{dongqif2, jingrui}@illinois.edu, {kaan}@gatech.edu,
{zhua, yanxie, fangjin, sizhang, haowu1, amalevich, bolong}@meta.com

## Abstract

Graph transformer has been proven as an effective graph learning method for its adoption of attention mechanism that is capable of capturing expressive representations from complex topological and feature information of graphs. Graph transformer conventionally performs dense attention (or global attention) for every pair of nodes to learn node representation vectors, resulting in quadratic computational costs that are unaffordable for large-scale graph data. Therefore, mini-batch training for graph transformers is a promising direction, but limited samples in each mini-batch can not support effective dense attention to encode informative representations. Facing this bottleneck, (1) we start by assigning each node a token list that is sampled by personalized PageRank (PPR) and then apply standard multi-head self-attention only on this list to compute its node representations. This PPR tokenization method decouples model training from complex graph topological information and makes heavy feature engineering offline and independent, such that mini-batch training of graph transformers is possible by loading each node's token list in batches. We further prove this PPR tokenization is viable as a graph convolution network with a fixed polynomial filter and jumping knowledge. However, only using personalized PageRank may limit information carried by a token list, which could not support different graph inductive biases for model training. To this end, (2) we rewire graphs by introducing multiple types of virtual connections through structure- and content-based super nodes that enable PPR tokenization to encode local and global contexts, long-range interaction, and heterophilous information into each node's token list, and then formalize our **V**irtual **C**onnection **R**anking based **Graph** Trans**former** (VCR-Graphormer). Overall, VCR-Graphormer needs $O(m+klogk)$ complexity for graph tokenization as compared to $O(n^3)$ of previous works. The code is provided [1].

## 1 Introduction

Transformer architectures have achieved outstanding performance in various computer vision and natural language processing tasks (Khan et al., 2021; Lin et al., 2022). Then, for non-grid graph data, developing effective graph transformers attracts much research attention, and some nascent works obtain remarkable performance in graph tasks for encoding complex topological and feature information into expressive node representations via attention mechanisms, like GT (Dwivedi and Bresson, 2020), Gophormer (Zhao et al., 2021), Coarformer (Kuang et al., 2021), Graphormer (Ying et al., 2021), SAN (Kreuzer et al., 2021), ANS-GT (Zhang et al., 2022), GraphGPS (Rampásek et al., 2022), NAGphormer (Chen et al., 2023), Exphormer (Shirzad et al., 2023) and etc.

Viewing each node as a token, most graph transformers need token-wise dense attention (or global attention) to extract expressive node embedding vectors (Müller et al., 2023), i.e., most graph transformers typically need to attend every pair of nodes in the input graph (Dwivedi and Bresson, 2020; Mialon et al., 2021; Ying et al., 2021; Kreuzer et al., 2021; Wu et al., 2021; Chen et al., 2022; Kim et al., 2022; Rampásek et al., 2022; Bo et al., 2023; Ma et al., 2023). As a consequence, dense attention induces unaffordable computational resource consumption for large-scale graph data due to this quadratic complexity, which hinders graph transformers from scaling up their effectiveness. In the meanwhile, mini-batch training provides a promising direction for graph transformers. Nonetheless,

---

a few node (or token) samples in each batch could not fully enable the dense attention mechanism to capture enough information for each node's representation vector, given the complex topological and feature information of the whole large graph data. NAGphormer (Chen et al., 2023) is a very recent work targeting mini-batch training, which utilizes self-attention only on multiple hop aggregations for a node's embedding vector and gets good experimental results. On the other side, the hop aggregation may not deal with global, long-range interactions, and heterophilous information well; also it relies on the time-consuming eigendecomposition for the positional encoding.

To deal with this bottleneck, first, we assign each target node a token list that is sampled by personalized PageRank (PPR). The token list consists of PPR-ranked neighbors with weights and will be self-attended by the standard transformer (Vaswani et al., 2017) to output the target node representation vector. This graph tokenization operation, based on PPR, separates the complex topological data from the model training process, allowing independent offline graph feature engineering for each node, such that the mini-batch training manner for graph transformers becomes possible by loading a few node token lists in each batch and leveraging self-attention separately. We demonstrate the feasibility of utilizing this graph tokenization-based graph transformer as an effective graph convolutional neural network, leveraging a fixed polynomial filter as described in (Kipf and Welling, 2017), and incorporating the concept of 'jumping knowledge' from (Xu et al., 2018). However, only using personalized PageRank to sample nodes can copy limited topological information into each token list, which could not support different graph inductive biases for model training. To this end, we propose to rewire the input graph by establishing multiple virtual connections for personalized PageRank to transfer local, global, long-range, and heterophilous information of the entire graph into each token list. Then, self-attention on this comprehensive token list enables a good approximation of the whole graph data during each batch of training. Wrapping personalized PageRank based graph tokenization and virtual connection based graph rewiring, we obtain an effective mini-batch graph transformer, called **V**irtual **C**onnection **R**anking based **Graph** Trans**former** (VCR-Graphormer). Comparing the $O(n^3)$ complexity of NAGphormer (Chen et al., 2023) due to the eigendecomposition, the complexity of VCR-Graphormer is $O(m + k log k)$, where $m$ is the number of edges in the graph and $k$ can be roughly interpreted as the selected neighbors for each node that is much smaller than $m$ or $n$.

## 2 PRELIMINARIES

**Notation**. We first summarize some common notations here, and detailed notions will be explained within context. Given an undirected and attributed graph $G = (V, E)$ with $|V| = n$ nodes and $|E| = m$ edges, we denote the adjacency matrix as $\mathbf{A} \in \mathbb{R}^{n \times n}$, the degree matrix as $\mathbf{D} \in \mathbb{R}^{n \times n}$, the node feature matrix as $\mathbf{X} \in \mathbb{R}^{n \times d}$, and the node label matrix as $\mathbf{Y} \in \mathbb{R}^{n \times c}$. For indexing, we denote $\mathbf{X}(v, :)$ as the row vector and $\mathbf{X}(:, v)$ as the column vector, and we use parenthesized superscript for index $\mathbf{X}^{(i)}$ and unparenthesized superscript for power operation $\mathbf{X}^i$.

**Self-Attention Mechanism**. According to Vaswani et al. (2017), the standard self-attention mechanism is a deep learning module that can be expressed as follows.

$$\mathbf{H}' = \text{Attention}(\mathbf{Q}, \mathbf{K}, \mathbf{V}) = \text{softmax}(\frac{\mathbf{Q}\mathbf{K}^\top}{\sqrt{d'}})\mathbf{V} \in \mathbb{R}^{d'} \tag{2.1}$$

where queries $\mathbf{Q} = \mathbf{H}\mathbf{W}_Q$, keys $\mathbf{K} = \mathbf{H}\mathbf{W}_K$ and values $\mathbf{V} = \mathbf{H}\mathbf{W}_V$. $\mathbf{H} \in \mathbb{R}^{n \times d}$ is the input feature matrix. $\mathbf{W}_Q, \mathbf{W}_K, \mathbf{W}_V \in \mathbb{R}^{d \times d'}$ are three learnable weight matrices. $\mathbf{H}' \in \mathbb{R}^{n \times d'}$ is the output feature matrix.

**Personalized PageRank**. Targeting a node of interest, the personalized PageRank vector stores the relative importance of the other existing nodes by exploring the graph structure through sufficient random walk steps. The typical personalized PageRank formula can be expressed as below.

$$\mathbf{r} = \alpha \mathbf{P} \mathbf{r} + (1 - \alpha)\mathbf{q} \tag{2.2}$$

where $\mathbf{P} \in \mathbb{R}^{n \times n}$ is the transition matrix, which can be computed as $\mathbf{A}\mathbf{D}^{-1}$ (column normalized (Yoon et al., 2018a)) or $\mathbf{D}^{-\frac{1}{2}}\mathbf{A}\mathbf{D}^{-\frac{1}{2}}$ (symmetrically normalized (Klicpera et al., 2019)). $\mathbf{q}$ is a one-hot stochastic vector (or personalized vector), and the element corresponding to the target node equals 1 while others are 0. $\mathbf{r}$ is the stationary distribution of random walks, i.e., personalized

PageRank vector, and $\alpha$ is the damping constant factor, which is usually set as 0.85 for personalized PageRank and 1 for PageRank. One typical way of solving a personalized PageRank vector is power iteration, i.e., multiplying the PageRank vector with the transition matrix by sufficient times $\mathbf{r}^{(t+1)} \longleftarrow \alpha \mathbf{P} \mathbf{r}^{(t)} + (1 - \alpha) \mathbf{q}$ until convergence or reaching the error tolerance.

## 3  EFFECTIVE MINI-BATCH TRAINING AND VCR-GRAPHORMER

In this section, we first propose a promising mini-batch way for graph transformer training and give theoretical analysis of its effectiveness. Then, we discuss its limitation and formalize an advanced version, **V**irtual **C**onnection **R**anking based **Graph** Trans**former** (VCR-Graphormer). Detailed proof of the proposed theory can be found in the appendix.

### 3.1  PPR TOKENIZATION FOR MINI-BATCH TRAINING

To project non-Euclidean topological information into expressive embedding vectors, most graph transformers need to use dense attention on every pair of nodes in the input graph. Set the $O(n^2)$ attention complexity aside, these graph transformers need to pay attention to every existing node that jeopardizes the mini-batch training for large-scale input graphs, because only a few samples in each batch are hard to allow dense attention to capture enough topological and feature information.

To achieve mini-batch training, graph topological and feature information should be effectively wrapped into each individual batch, and then the graph learning model can start to learn on each batch. Thus, an inevitable way is to first disentangle the message passing along the topology before the model training and place the disentanglement properly in each batch. Motivated by this goal, we propose a mini-batch training paradigm based on personalized PageRank (PPR) tokenization. The logic of this paradigm is easy to follow, which tokenizes the input graph by assigning each target node $u$ with a token list $\mathcal{T}_u$. This list $\mathcal{T}_u$ consists of tokens (nodes) that are related to the target node $u$. Here, we use personalized PageRank (PPR) to generate the token list. Set node $u$ as the target, based on Eq. 2.2, we can get the personalized PageRank vector, denoted as $\mathbf{r}_u$. Then, we sample the top $k$ nodes in $\mathbf{r}_u$, denoted as $\mathcal{R}_u^k$. Basically, nodes in $\mathcal{R}_u^k$ together form the token list $\mathcal{T}_u$. Yet, the list $\mathcal{T}_u$ will feed to the transformer (Vaswani et al., 2017), then the positional encoding is necessary. Otherwise, two target nodes that share the same PPR-based neighbors but differ in ranking orders will result in the identical representation vector by self-attention. Actually, PPR ranking scores (i.e., the probability mass stored in the PPR vector $\mathbf{r}_u$) can provide good positional encoding information. Moreover, the PPR ranking scores are personalized, which means two token lists that share exactly the same entries can be discriminated by different PPR ranking scores from different target nodes. Mathematically, our general $\mathcal{T}_u$ is more expressed as aggregated form $\mathcal{T}_u^{Agg}$ and discrete form $\mathcal{T}_u^{Dis}$ as follows.

$$\mathcal{T}_u^{Agg} = \{(\mathbf{P}^l \mathbf{X})(u,:)\}, \text{ s.t. for } l \in \{1, \dots, L\} \tag{3.1}$$

where $\mathbf{X} \in \mathbb{R}^{n \times d}$ is the feature matrix, and $\mathbf{P}^l \mathbf{X}$ is the $l$-th step random walk. At the $l$-th step, if we decompose the summarization, the discrete form is expressed as follows.

$$\mathcal{T}_u^{Dis} = \{\mathbf{X}(i,:) \cdot \mathbf{r}_u(i)\}, \text{ s.t. for } i \in \mathcal{R}_u^k \tag{3.2}$$

where $\mathbf{r}_u$ is the (personalized) PageRank vector at the $l$-th step starting from node $u$, and $\mathcal{R}_u^k$ is the set containing the top $k$ entries from $\mathbf{r}_u$.

Based on Eq. 3.1 (or Eq. 3.2), we can reorganize the set $\mathcal{T}_u$ into a matrix $\mathbf{T}_u \in \mathbb{R}^{l \times d}$ in the aggregated form (or $\mathbf{T}_u \in \mathbb{R}^{lk \times d}$ in the discrete form) by stacking entries. Then, we can input $\mathbf{T}_u$ into the standard self-attention mechanism stated in Eq. 2.1 and a pooling function (e.g., mean or sum) to get the representation vector $\mathbf{h}_u$ for node $u$. This kind of graph tokenization with the standard self-attention mechanism is viable for approximating a graph convolution neural network as stated in Theorem 3.1.

**Theorem 3.1** *Sampling token lists through (personalized) PageRank and using token-wise self-attention on each token list, this operation is equivalent to a fixed polynomial filter graph convolution operation (Kipf and Welling, 2017) with jumping knowledge Xu et al. (2018).*

As stated in Theorem 3.1, the graph tokenization operation in Eq. 3.1 is viable for approximating GCN (Kipf and Welling, 2017), but it is prone to have the latent incapabilities of GCN, such like

Table 1: Four Metrics from Inductive Bias and Efficiency for Graph Tokenization Methods

| A Good Graph Tokenization Could | Hop2Token (Chen et al., 2023) | Personalized PageRank | VCR-Graphormer |
|---|---|---|---|
| 1. Reflect the input graph topology (local and global) | 🙂 | 😐 | 🙂 |
| 2. Support long-range interaction | 😐 | 🙁 | 🙂 |
| 3. Handle heterophily property | 🙁 | 🙁 | 🙂 |
| 4. Be efficiently realized | 😐 | 🙂 | 🙂 |

shallow network or low-pass filter can only preserve local and homophilous information (Bo et al., 2021) but the deeper network can suffer from over-squashing problem (Topping et al., 2022). Hence, the next direction is how to warp good input graph information into the token list to alleviate the aforementioned problems.

## 3.2 PRINCIPLES OF A "GOOD" TOKENIZATION

To solve the problems mentioned in the above subsection, in general, a good token list should contain enough information that can be generalizable to different graph inductive biases, whereas graph inductive bias refers to the assumptions or preconceptions that a machine learning model makes about the underlying distribution of graph data. Because we have compressed the entire graph topology and features (that are easily accessible for each node in the dense or global attention graph transformers (Dwivedi and Bresson, 2020; Mialon et al., 2021; Ying et al., 2021; Kreuzer et al., 2021; Wu et al., 2021; Chen et al., 2022; Kim et al., 2022; Rampásek et al., 2022; Bo et al., 2023; Ma et al., 2023)) into the token list for the mini-batch training, we need to enable a good approximation of the whole graph data in each sampled token list. By attending to this good approximation, the self-attention mechanism can produce representative node embeddings. To define the goodness of the sampled token list, we propose at least four principles that need to be satisfied.

**First**, the token list of each node should reflect the local and global topological information of the input graph. For instance, starting from a target node, personalized PageRank can sample the top neighbor nodes, whose ranking can be regarded as local topology (Andersen et al., 2006). Also, if we eigendecompose the graph Laplacian matrix, the eigenvalues and eigenvectors can preserve the global topological information (Spielman, 2007). **Second**, the information in the token list should preserve long-range interactions of the input graph. The long-range interaction problem refers to the expressive node representation sometimes relying on a relatively long-distance message passing path in GNNs (Alon and Yahav, 2021; Dwivedi et al., 2022; Yan et al., 2023b), and just stacking more layers in GNNs is found not solving the long-range interaction need but inducing over-squashing problem due to the compressed receptive field for each node (Alon and Yahav, 2021; Topping et al., 2022; Karhadkar et al., 2023). **Third**, the information in the token list should have the ability to contain possible heterophilous information of the graph. The heterophily property of graphs can be interpreted as the close neighbors (e.g., connected nodes) not sharing similar content (e.g., node labels) (Bo et al., 2021; Platonov et al., 2023; Yan et al., 2023a), such that structure-based message passing in GNNs alone may not fully meet downstream tasks when the topological distribution and feature distribution do not conform. **Fourth**, the token list should be efficiently obtained. For example, although eigendecomposition preserves graph topology, it needs $O(n^3)$ computational complexity (Shi and Malik, 1997). In general, the design of graph tokenization for graph transformers should avoid the problems discovered previously in GNNs. To be more specific, the list should be efficiently obtained but contain comprehensive information to provide sufficient graph inductive biases for model training.

The four metrics are listed in Table 1, where we then analyze if the related works satisfy the requirements. The nascent graph transformer NAGphormer (Chen et al., 2023) proposes the Hop2Token method, which first performs eigendecomposition of the Laplacian matrix of the input graph and attaches the eigenvector to each node input feature vector. Then, for a target node, the summarization of the 1-hop neighbors' feature vectors serves as the first entry in the target node's token list. Finally, NAGphormer uses self-attention on the $k$-length token list to output the representation vector for the target node. Due to the $k$-hop aggregation and eigendecomposition, Hop2Token could preserve the local and global topological information of the input graph. But for the long-range interaction aware-

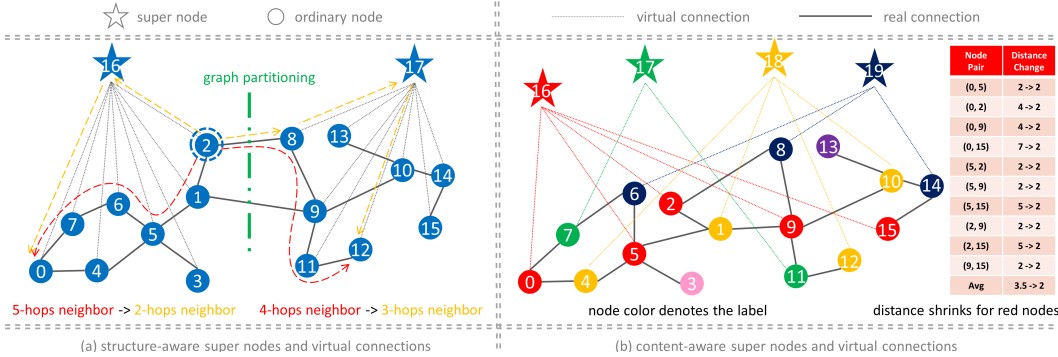

Figure 1: (a) Structure-Aware and (b) Content-Aware Virtual Connections on the Same Input Graph.

ness, hop-based aggregation may diminish node-level uniqueness, and computing the polynomial of the adjacency matrix is costly when $k$ needs to be considerably large. Moreover, Hop2Token pays attention to structurally close neighbors and ignores possible heterophily properties of graphs. Also, additional eigendecomposition costs cubic complexity to obtain higher performance. Personalized PageRank (as discussed in Sec.3.1) samples local neighbors for a target node and overlooks global contexts, also it can not support long-range interaction when necessary. Although it ignores the heterophily, the personalized ranking list can be obtained in a constant time complexity (Andersen et al., 2006). Our proposed VCR-Graphormer can satisfy these four metrics in its sampled token list. Next, we introduce it with theoretical analysis.

## 3.3 VCR-GRAPHORMER

Here, we first introduce how the token list of each node in VCR-Graphormer gets formed, and then we give the theoretical analysis. In general, the token list of VCR-Graphormer consists of four components. For a target node $u$, the token list $\mathcal{T}_u$ is expressed as follows.

$$\mathcal{T}_u = \{\mathbf{X}(u,:)||1,\ (\mathbf{P}^l\mathbf{X})(u,:)||\frac{L-l+1}{\sum_{l=1}^{L} l},\ \mathbf{X}(i,:)||\bar{\mathbf{r}}_u(i),\ \mathbf{X}(j,:)||\hat{\mathbf{r}}_u(j)\},$$
$$\text{s.t. for } l \in \{1,\dots,L\}, i \in \bar{\mathcal{R}}_u^{\bar{k}}, j \in \hat{\mathcal{R}}_u^{\hat{k}} \tag{3.3}$$

where these four components have different views of the input graph, and each component has an initial scalar weight, i.e., 1. Also, $|\mathcal{T}_u| = 1 + L + \bar{k} + \hat{k}$. We then introduce details of the four components. $\{\}$ operator means a list. The concatenation operator applies to both scalars and vectors, i.e., the scalar will be converted to a single-element vector before concatenation.

The **first component**, $\mathbf{X}(u,:)||1 \in \mathbb{R}^{d+1}$ is straightforward, which is the input feature of the target node $u$ associated with the weight 1. The **second component** is designed to capture the local topological information around the target node $u$, based on the aggregated form of PPR sampling as expressed in Eq 3.1. Matrix $\mathbf{P}^l\mathbf{X}$ is the $l$-th step of random walks, since one more random walk step accesses 1-hop away neighbors (Klicpera et al., 2019), those neighbors' features are aggregated based on random walk probability distribution to encode the local topological information for node $u$. Moreover, the given weight 1 for $\mathbf{P}^l\mathbf{X}$ is distributed among walk steps by $\frac{L-l+1}{\sum_{l=1}^{L} l}$, which means the closer neighbors' aggregation occupies more weights.

So far, the first two components are proposed to meet the half part of the first metric in Table 1, i.e., the local graph topology gets preserved in the token list. After that, the third component is responsible for recording long-range interactions and partial global information (i.e., from the node connections perspective), and the fourth component is designed to record the heterophily information and partial global information (i.e., from the node content perspective).

The **third component** is motivated by graph rewiring, but our goal is to let the token list preserve structure-based global information and extend the narrow receptive field of over-squashing (Alon and Yahav, 2021) in an efficient manner. To be specific, we break the original structure by inserting several virtual **super nodes** into the graph adjacency matrix. To differentiate from the inserted super

nodes, the originally existing nodes are named **ordinary nodes**. To do this, the input graph $G$ is first partitioned into $\bar{s}$ clusters (e.g., using METIS partitioning algorithm (Karypis and Kumar, 1998)). For each cluster, we assign a super node that connects every member in this cluster, where the new edges are called structure-aware virtual connections, as shown in Figure 1(a). After that, the original adjacency matrix is transferred from $\mathbf{A} \in \mathbb{R}^{n \times n}$ to $\bar{\mathbf{A}} \in \mathbb{R}^{(n+\bar{s}) \times (n+\bar{s})}$. Note that, in $\bar{\mathbf{A}}$, the inter-cluster edges are retained. Correspondingly, the transition matrix for PPR is denoted as $\bar{\mathbf{P}}$. Then, based on $\bar{\mathbf{P}}$, we can get the personalized PageRank vector $\bar{\mathbf{r}}_u$ through Eq. 2.2. Again, we sample top $\bar{k}$ items in $\bar{\mathbf{r}}_u$ to form the set $\bar{\mathcal{R}}_u^{\bar{k}}$. Finally, we can obtain the complete vector of the third component, i.e., $\mathbf{X}(i,:) || \bar{\mathbf{r}}_u(i) \in \mathbb{R}^{d+1}$, for each node $i \in \bar{\mathcal{R}}_u^{\bar{k}}$. Compared with the second component, we adapt the discrete form as stated in Eq. 3.2 for the third component for the following two reasons. (1) We assume the long-range interaction between nodes is represented in the form of paths other than broadcasts. Then, still concatenating all nodes from far-away hops can induce noise, but personalized PageRank is more suitable for sampling paths with corresponding weights. (2) Multiplying matrices $\mathbf{PX}$ as the second component will at least cost $O(m)$ time complexity, which is linear to the number of edges. On the contrary, personalized PageRank can be completed in a constant time, which is independent of the input graph size (Andersen et al., 2006). The above two reasons also apply to the design of the following fourth component.

The **fourth component** is designed to capture the content-based global information and encode the heterophily information into the token list. The content here refers to the labels of nodes or pseudo-labels like feature clustering and prototypes (Snell et al., 2017) when labels are not available. The realization of the fourth component is quite similar to the third component. The difference is that a super node here is assigned to one kind of label in the input graphs, and then it connects every node that shares this label. Those new edges are called content-aware virtual connections, as shown in Figure 1(b). Note that, when the input training set has labels, the number of content-aware super nodes $\hat{s}$ equals the number of node labels $c$, otherwise it is a hyperparameter, e.g., feature-based clusters in $K$-means. Also, adding content-aware super nodes is based on the original adjacency matrix, i.e., $\mathbf{A} \in \mathbb{R}^{n \times n}$ to $\hat{\mathbf{A}} \in \mathbb{R}^{(n+\hat{s}) \times (n+\hat{s})}$. Then, given $\hat{\mathbf{P}}$, the procedure of getting $\hat{\mathbf{r}}_u$, $\hat{\mathcal{R}}_u^{\hat{k}}$, and $\mathbf{X}(j,:) || \hat{\mathbf{r}}_u(j) \in \mathbb{R}^{d+1}$ for $j \in \hat{\mathcal{R}}_u^{\hat{k}}$, is identical to the third component.

After stacking elements in the set $\mathcal{T}_u$, we get the matrix $\mathbf{T}_u \in \mathbb{R}^{(1+L+\bar{k}+\hat{k}) \times (d+1)}$ as the input of the standard transformer, i.e., for node $u$, the initial input $\mathbf{Z}_u^{(0)} = \mathbf{T}_u$. Then, for the $t$-th layer,

$$\mathbf{Z}_u^{(t)} = \text{FFN}(\text{LN}(\tilde{\mathbf{Z}}_u^{(t)})) + \tilde{\mathbf{Z}}_u^{(t)}, \quad \tilde{\mathbf{Z}}_u^{(t)} = \text{MHA}(\text{LN}(\mathbf{Z}_u^{(t-1)})) + \mathbf{Z}_u^{(t-1)} \tag{3.4}$$

where LN stands for layer norm, FFN is feedforward neural network, and MHA denotes multi-head self-attention. To get the representation vector for node $u$, a readout function (e.g., mean, sum, attention) will be applied on $\mathbf{Z}_u^{(t)}$. We give the analysis of our proposed techniques in Appendix.

# 4 EXPERIMENTS

## 4.1 SETUP

**Datasets**. Totally, we have 13 publicly available graph datasets included in this paper. First, as shown in Table 2, 9 graph datasets are common benchmarks for node classification accuracy performance, 6 of them are relatively small, and 3 of them are large-scale datasets. Small datasets like PubMed, CoraFull, Computer, Photo, CS, and Physics can be accessed from the DGL library [2]. Reddit, Aminer, and Amazon2M are from (Feng et al., 2022) can be accessed by this link [3]. Second,

Table 2: Graph Datasets and Statistics

|          | PubMed | CoraFull | Computer | Photo | CS | Physics | Reddit | Aminer | Amazon2M |
|----------|--------|----------|----------|-------|-----|---------|--------|--------|----------|
| Nodes    | 19,717 | 19,793   | 13,752   | 7,650 | 18,333 | 34,493 | 232,965 | 593,486 | 2,449,029 |
| Edges    | 88,651 | 126,842  | 491,722  | 238,163 | 163,788 | 495,924 | 11,606,919 | 6,217,004 | 61,859,140 |
| Features | 500    | 8,710    | 767      | 745   | 6,805 | 8,415   | 602    | 100    | 100      |
| Labels   | 3      | 70       | 10       | 8     | 15  | 15      | 41     | 18     | 47       |

we also have 3 small heterophilous graph datasets, Squirrel, Actor, and Texas, where the connected

---

[2] https://docs.dgl.ai/api/python/dgl.data.html
[3] https://github.com/THUDM/GRAND-plus

neighbors mostly do not share the same label (i.e., measured by node heterophily in (Zhang et al., 2023)). They can also be accessed from the DGL library. For small-scale datasets and heterophilous datasets, we apply 60%/20%/20% train/val/test random splits. For large-scale datasets, we follow the random splits from (Feng et al., 2022; Chen et al., 2023). Third, we also include the large-scale heterophious graph dataset, arXiv-Year, from the benchmark (Lim et al., 2021).

**Baselines**. Our baseline algorithms consist of 4 categories. For GNNs, we have GCN (Kipf and Welling, 2017), GAT (Velickovic et al., 2018), APPNP (Klicpera et al., 2019), and GPR-GNN (Chien et al., 2021). For scalable GNNs, we include GraphSAINT (Zeng et al., 2020), PPRGo (Bojchevski et al., 2020), and GRAND+ (Feng et al., 2022). For graph transformers, we

Table 3: Heterophilous Graph Datasets and Statistics

|  | Squirrel | Actor | Texas |
|---|---|---|---|
| Nodes | 5,201 | 7,600 | 183 |
| Edges | 216,933 | 33,544 | 295 |
| Features | 2,089 | 931 | 1,703 |
| Labels | 5 | 5 | 5 |
| Node Heterophily | 0.78 | 0.76 | 0.89 |

have GT (Dwivedi and Bresson, 2020), Graphormer (Ying et al., 2021), SAN (Kreuzer et al., 2021), and GraphGPS (Rampásek et al., 2022). As for scalable graph transformers, we have NAGPhormer (Chen et al., 2023) discussed above, and Exphormer (Shirzad et al., 2023) is a new scalable graph transformer, which also needs relative dense attention. Additionally, we also include various baselines from (Lim et al., 2021) as shown in Appendix.

## 4.2 PERFORMANCE ON DIFFERENT SCALE DATASETS

The node classification accuracy on small-scale graph datasets and large-scale datasets among various baseline algorithms is reported in Table 4 and Table 5 respectively. Containing more structure-aware neighbors (i.e., larger $\bar{k}$) and more content-aware neighbors (i.e., larger $\hat{k}$) in the token list has the potential to increase the node classification performance. For attention efficiency, we constrain $\bar{k}$ and $\hat{k}$ to be less than 20 in VCR-Graphormer, even for large-scale datasets.

Table 4: Node Classification Accuracy on Small-Scale Datasets

|  |  | PubMed | CoraFull | Computer | Photo | CS | Physics |
|---|---|---|---|---|---|---|---|
| GNN | GCN | $86.54 \pm 0.12$ | $61.76 \pm 0.14$ | $89.65 \pm 0.52$ | $92.70 \pm 0.20$ | $92.92 \pm 0.12$ | $96.18 \pm 0.07$ |
|  | GAT | $86.32 \pm 0.16$ | $64.47 \pm 0.18$ | $90.78 \pm 0.13$ | $93.87 \pm 0.11$ | $93.61 \pm 0.14$ | $96.17 \pm 0.08$ |
|  | APPNP | $88.43 \pm 0.15$ | $65.16 \pm 0.28$ | $90.18 \pm 0.17$ | $94.32 \pm 0.14$ | $94.49 \pm 0.07$ | $96.54 \pm 0.07$ |
|  | GPR-GNN | $89.34 \pm 0.25$ | $67.12 \pm 0.31$ | $89.32 \pm 0.29$ | $94.49 \pm 0.14$ | $95.13 \pm 0.09$ | $96.85 \pm 0.08$ |
| Scalable GNN | GraphSAINT | $88.96 \pm 0.16$ | $67.85 \pm 0.21$ | $90.22 \pm 0.15$ | $91.72 \pm 0.13$ | $94.41 \pm 0.09$ | $96.43 \pm 0.05$ |
|  | PPRGo | $87.38 \pm 0.11$ | $63.54 \pm 0.25$ | $88.69 \pm 0.21$ | $93.61 \pm 0.12$ | $92.52 \pm 0.15$ | $95.51 \pm 0.08$ |
|  | GRAND+ | $88.64 \pm 0.09$ | $71.37 \pm 0.11$ | $88.74 \pm 0.11$ | $94.75 \pm 0.12$ | $93.92 \pm 0.08$ | $96.47 \pm 0.04$ |
| Graph Transformer | GT | $88.79 \pm 0.12$ | $61.05 \pm 0.38$ | $91.18 \pm 0.17$ | $94.74 \pm 0.13$ | $94.64 \pm 0.13$ | $97.05 \pm 0.05$ |
|  | Graphormer | OOM | OOM | OOM | $92.74 \pm 0.14$ | $94.64 \pm 0.13$ | OOM |
|  | SAN | $88.22 \pm 0.15$ | $59.01 \pm 0.34$ | $89.93 \pm 0.16$ | $94.86 \pm 0.10$ | $94.51 \pm 0.15$ | OOM |
|  | GraphGPS | $88.94 \pm 0.16$ | $55.76 \pm 0.23$ | OOM | $95.06 \pm 0.13$ | $93.93 \pm 0.15$ | OOM |
| Scalable Graph Transformer | NAGphormer | $89.70 \pm 0.19$ | $71.51 \pm 0.13$ | $91.22 \pm 0.14$ | $95.49 \pm 0.11$ | $95.75 \pm 0.09$ | $\mathbf{97.34 \pm 0.03}$ |
|  | Exphormer | $89.52 \pm 0.54$ | $69.09 \pm 0.72$ | $91.59 \pm 0.31$ | $95.27 \pm 0.42$ | $\mathbf{95.77 \pm 0.15}$ | $97.16 \pm 0.13$ |
|  | VCR-Graphormer | $\mathbf{89.77 \pm 0.15}$ | $\mathbf{71.67 \pm 0.10}$ | $\mathbf{91.75 \pm 0.15}$ | $\mathbf{95.53 \pm 0.14}$ | $95.37 \pm 0.04$ | $\mathbf{97.34 \pm 0.04}$ |

From Table 4, our VCR-Graphormer shows the competitive performance among all baseline methods. Moreover, VCR-Graphormer does not need the time-consuming eigendecomposition as another scalable graph transformer NAGPhormer Chen et al. (2023), and the corresponding structure-aware neighbors and content-aware neighbors via virtual connections can help preserve the global information. For Table 5, VCR-Graphormer also achieves very competitive performance with limited token length, which also demonstrates that several sampled global neighbors may not bridge the effectiveness of eigendecomposition.

## 4.3 PERFORMANCE ON HETEROPHILOUS DATASETS

Since heterophilous datasets are small, we only allow VCR-Graphormer to take less than 10 global neighbors for Squirrel and Actor datasets, and less than 5 for Texas dataset. In Figure 2, the $x$-axis denotes how many hops' information can be aggregated and leveraged for the baseline algorithm. The performance in Figure 2 demonstrates the analysis in Table 1, that hop-aggregation and eigen-

Table 5: Node Classification Accuracy on Large-Scale Datasets

|  |  | Reddit | Aminer | Amazon2M |
|---|---|---|---|---|
| Scalable GNN | PPRGo | $90.38 \pm 0.11$ | $49.47 \pm 0.19$ | $66.12 \pm 0.59$ |
|  | GraphSAINT | $92.35 \pm 0.08$ | $51.86 \pm 0.21$ | $75.21 \pm 0.15$ |
|  | GRAND+ | $92.81 \pm 0.03$ | $\mathbf{54.67 \pm 0.25}$ | $75.49 \pm 0.11$ |
| Scalable Graph Transformer | NAGphormer | $93.58 \pm 0.05$ | $54.04 \pm 0.16$ | $\mathbf{77.43 \pm 0.11}$ |
|  | Exphormer | OOM | NA4HETERO | OOM |
|  | VCR-Graphormer | $\mathbf{93.69 \pm 0.08}$ | $53.59 \pm 0.10$ | $76.09 \pm 0.16$ |

decomposition could not preserve the latent heterophily information very well, and content-aware neighbors though virtual connections can alleviate this problem.

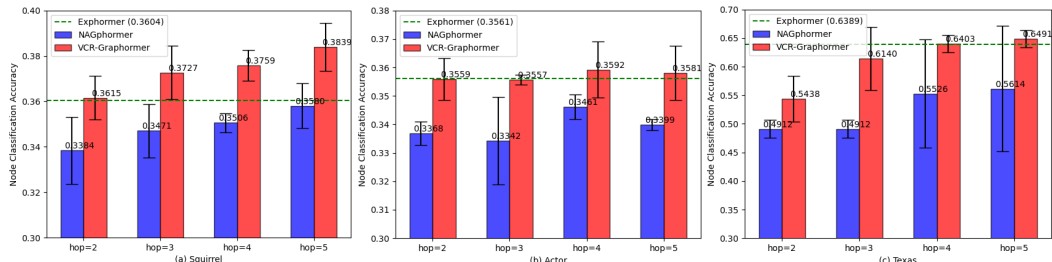

Figure 2: Node Classification Accuracy on Heterophilous Datasets

## 4.4 ABLATION STUDY AND PARAMETER ANALYSIS

After showing the performance above, we would verify the following research questions. RQ1: Does structure-aware neighbors and content-aware neighbors contribute to the performance? RQ2: Are the meaning and function of structure-aware neighbors and content-aware neighbors different and complementary? RQ3: What is the relationship among the number of structure-aware super nodes, the number of hops, and the number of structure- and content-aware neighbors?

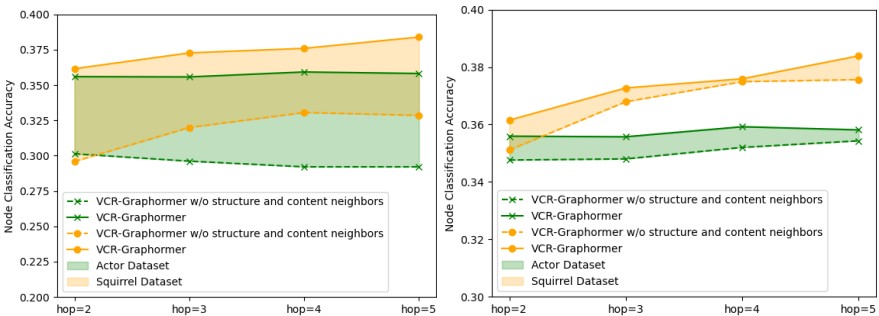

Figure 3: Ablation Study on Squirrel and Actor

For RQ1 and RQ2, we need to investigate the role of structure- and content-aware neighbors. The heterophilous datasets are a good fit. For example, we can see from Figure 3, not sampling structure- and content-aware neighbors (i.e., (0, 0)) or only sampling structure-aware neighbors (i.e., (10, 0)) could not achieve the performance as having both (i.e., (10, 10)). For RQ3, we select the large-scale dataset, Reddit. From Figure 4, if we decrease $l$, we lose the local information and performance reduces from 93.36 to 93.02. To mitigate this local information, we need to shrink the scope of a structure-aware super node (i.e., increasing $\bar{s}$) and sample more neighbors, the performance increases from 93.02 to 93.16.

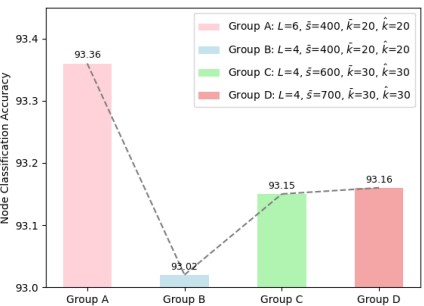

Figure 4: Parameter Analysis on Reddit

## 5 RELATED WORK

Graph data and representation learning techniques have a wide scope of applications like recommendation (Qi et al., 2023; Ban et al., 2022; Wei et al., 2021), query answering (Li et al., 2023; Liu et al., 2021; Yan et al., 2021), misinformation detection (Fu et al., 2022a), time series analysis (Jing et al., 2021), protein classification and generation (Fu et al., 2022b; Zhou et al., 2022), etc. Recently, graph transformers emerged as one kind of effective graph representation learning model (Müller et al., 2023), which adapts the self-attention mechanism from transformers (Vaswani et al., 2017) to graph data representation learning. For example, GT (Dwivedi and Bresson, 2020) views each node in the input graph as a token and uses the self-attention mechanism to attend every pair of nodes to get the representation vectors. Then, various graph transformers are proposed following the dense attention mechanism with different positional encodings for node tokens (Mialon et al., 2021; Ying et al., 2021; Kreuzer et al., 2021; Wu et al., 2021; Chen et al., 2022; Kim et al., 2022; Rampásek et al., 2022; Bo et al., 2023; Ma et al., 2023), such that the standard transformer can learn topological information during the model training. For example, in TokenGT (Kim et al., 2022), each node token is assigned augmented features from orthogonal random features and Laplacian eigenvectors, such that the edge existence can be identified by the dot product of two token vectors. Also, GraphTrans (Wu et al., 2021) connects a GNN module with the standard transformer and utilizes the GNN to provide topology-preserving features to the self-attention mechanism. To reduce the quadratic attention complexity of dense attention (or global attention), some recent works propose approximation methods (Rampásek et al., 2022; Shirzad et al., 2023; Kong et al., 2023). In GraphGPS (Rampásek et al., 2022), the linear attention layer of Performer (Choromanski et al., 2021) is proposed to replace the standard transformer layer in graph transformers but the performance gain is marginal, where the authors hypothesized that the long-range interaction is generally important and cannot be easily captured in the linear attention mechanism. Exphormer (Shirzad et al., 2023) proposes using sparse attention in graph transformers, which intentionally selects several edges (e.g., by establishing expander graphs) to execute the node-wise attention. To address the quadratic complexity, GOAT (Kong et al., 2023) follows the dense attention but targets to reduce the feature dimension by finding a proper projection matrix, in order to save the efficiency. Among the above, if not all, most graph transformers are designed for graph-level tasks like graph classification and graph regression (Hu et al., 2020). Targeting node-level tasks like node classification, some works are deliberately proposed (Wu et al., 2022; Chen et al., 2023). For example, NAGphormer Chen et al. (2023) is proposed to assign each node a token list, and then the self-attention mechanism is only needed for this list to get the node embedding instead of handling all existing node pairs. This paradigm promises scalability for mini-batch training of graph transformers. Based on that, in this paper, we first show why this paradigm can work well, propose the criteria of a good token list for carrying various graph inductive biases, and propose the efficient and effective graph transformer, VCR-Graphormer.

## 6 CONCLUSION

In this paper, we propose the PageRank based node token list can support graph transformers with mini-batch training and give the theoretical analysis that the token list based graph transformer is equivalent to a GCN with fixed polynomial filter and jumping knowledge. Then, we propose criteria for constructing a good token list to encode enough graph inductive biases for model training. Based on the criteria, we propose the corresponding techniques like virtual connections to form a good token list. Finally, the proposed techniques are wrapped into an end-to-end graph transformer model, VCR-Graphormer, which can achieve outstanding effectiveness in an efficient and scalable manner.

## ACKNOWLEDGEMENT

This work was partially done when the first author was a research scientist intern at Meta AI. This work was also partially supported by National Science Foundation under Award No. IIS-2117902, and the U.S. Department of Homeland Security under Grant Award Number, 17STQAC00001-06-00. The views and conclusions are those of the authors and should not be interpreted as representing the official policies of the funding agencies or the government.

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

# APPENDIX

## 6.1 PROOF OF THEOREM 3.1

According to (Kipf and Welling, 2017), given an undirected graph $G$ with $n$ nodes, the corresponding graph convolution operation on a signal matrix $\mathbf{X} \in \mathbb{R}^{n \times d}$ is defined as

$$\mathbf{U} g_\theta(\mathbf{\Lambda}) \mathbf{U}^\top \mathbf{X} \approx \mathbf{U}(\sum_{k=0}^{K} \theta_k \mathbf{\Lambda}^k) \mathbf{U}^\top \mathbf{X} \tag{6.1}$$

where $\mathbf{L} = \mathbf{I} - \mathbf{D}^{-\frac{1}{2}} \mathbf{A} \mathbf{D}^{-\frac{1}{2}} \in \mathbb{R}^{n \times n}$ is the normalized Laplacian matrix, $\mathbf{A} \in \mathbb{R}^{n \times n}$ is the adjacency matrix, and $\mathbf{D} \in \mathbb{R}^{n \times n}$ is the diagonal degree matrix. Then, $\mathbf{U} \in \mathbb{R}^{n \times n}$ is the matrix containing eigenvectors of Laplacian matrix $\mathbf{L}$ and $\mathbf{\Lambda} \in \mathbb{R}^{n \times n}$ is the diagonal matrix storing the corresponding eigenvalues.

Based on $(\mathbf{U}\mathbf{\Lambda}\mathbf{U}^\top)^k = \mathbf{U}\mathbf{\Lambda}^k\mathbf{U}^\top$, extending the above equation can get

$$\mathbf{U} g_\theta(\mathbf{\Lambda}) \mathbf{U}^\top \mathbf{X} \approx (\sum_{k=0}^{K} \theta_k \mathbf{L}^k) \mathbf{X} \tag{6.2}$$

Then, Kipf and Welling (2017) approximately take $K = 1$, $\theta_0 = 2\theta$, and $\theta_1 = -\theta$, obtain

$$\mathbf{U} g_\theta(\mathbf{\Lambda}) \mathbf{U}^\top \mathbf{X} \approx \theta(\mathbf{I} + \mathbf{D}^{-\frac{1}{2}} \mathbf{A} \mathbf{D}^{-\frac{1}{2}}) \mathbf{X} \tag{6.3}$$

Finally, replacing $\mathbf{I} + \mathbf{D}^{-\frac{1}{2}} \mathbf{A} \mathbf{D}^{-\frac{1}{2}}$ above by $\mathbf{P} = (\mathbf{D} + \mathbf{I})^{-\frac{1}{2}} (\mathbf{A} + \mathbf{I})(\mathbf{D} + \mathbf{I})^{-\frac{1}{2}}$, the final format of graph convolution operation is

$$\mathbf{H}^{(t+1)} = \sigma(\mathbf{P}\mathbf{H}^{(t)}\mathbf{\Theta}^{(t)}) \tag{6.4}$$

where $t$ is the index of the graph convolution layer, $\mathbf{H}^0 = \mathbf{X}$, and $\sigma$ is a non-linear activation function.

Removing the non-linear activation function at each intermediate layer (Wu et al., 2019), the final representation output $\mathbf{Z}$ of a $T$-layer graph convolution neural network can be rewritten as

$$\mathbf{Z} = \sigma(\mathbf{P}^T \mathbf{X} \boldsymbol{\Theta}^{(1)}, \dots, \boldsymbol{\Theta}^{(T)}) = \sigma(\mathbf{P}^T \mathbf{X} \boldsymbol{\Theta}) \tag{6.5}$$

which shows the above-approximated equation is a GCN with a fixed polynomial filter, which is also the second component of our Eq. 3.3. The only difference is that, the second component of Eq. 3.3 keeps multiple $t \in \{1, \dots, T\}$, i.e., $[\mathbf{P}^1 \mathbf{X}, \mathbf{P}^2 \mathbf{X}, \dots, \mathbf{P}^T \mathbf{X}]$, which are corresponding to Jumping Knowledge (Xu et al., 2018) to learn representations of different orders for different graph substructures through the attention pooling.

Moreover, for personalized PageRank, $\mathbf{P}^t \mathbf{X}$ can be replaced as $\alpha(\mathbf{I} - (1 - \alpha)(\mathbf{A} + \mathbf{I}))^{-1}\mathbf{X}$, if $t$ is sufficiently large for convergence; otherwise, it can be obtained iteratively by $\mathbf{H}^{(t+1)} = (1 - \alpha)\mathbf{P}\mathbf{H}^{(t)} + \alpha\mathbf{X}$, where $\mathbf{H}^{(0)} = \mathbf{X}$, which is

$$(1 - \alpha)^t \mathbf{P}^t \mathbf{X} + (1 - \alpha)^{t-1} \mathbf{P}^{t-1} \mathbf{X} + \dots + (1 - \alpha)^1 \mathbf{P}^1 \mathbf{X} + \alpha\mathbf{X} \tag{6.6}$$

## 6.2 THEORITICAL ANALYSIS OF VCR-GRAPHORMER

**Remark 6.1** *After inserting (structure- or content-aware) super nodes, towards a target ordinary node, the remaining ordinary nodes can be classified into two categories. The first category consists of the nodes whose shortest path to the target node gets decreased. The second category consists of the nodes whose shortest path to the target node remains. We name the first category as **brightened ordinary nodes** and the second category as **faded ordinary nodes**. Faded ordinary nodes denote the local neighbors of the target node, because they are originally two-hop accessible nodes. Brightened ordinary nodes consist of the remaining ordinary nodes in the graph.*

**Theorem 6.1** *After inserting (structure- or content-aware) super nodes and changing the original transition matrix $\mathbf{P}$ into new $\tilde{\mathbf{P}}$, the probability mass of personalized (e.g., starting from node u) PageRank transforming from faded ordinary nodes to brightened ordinary nodes is the summation of all negative entries in the vector $\mathbf{c} = \alpha \sum_i^\infty \alpha^i \tilde{\mathbf{P}}^i (\tilde{\mathbf{P}} - \mathbf{P})\mathbf{r}$, where $\mathbf{r}$ is the PageRank vector on the original graph $\mathbf{P}$. Correspondingly, all the positive entries are received mass for brightened ordinary nodes. Vector $\mathbf{c}$ will converge to a constant vector and the sum of vector $\mathbf{c}$ is 0.*

The general insights from Theorem 6.1 show the probability mass transfer from faded ordinary nodes to brightened ordinary nodes, which further illustrates that previous hardly-reachable nodes are obtaining more weight to extend the narrow receptive field.

### 6.2.1 PROOF OF THEOREM 6.1

We first introduce a solution for exactly solving personalized PageRank vector, called cumulative power iteration (Yoon et al., 2018b), which is expressed as follows.

$$\mathbf{r} = \sum_{i=0}^{\infty} \mathbf{c}^{(i)}, \quad \mathbf{c}^{(i)} = (\alpha\mathbf{P})^i (1 - \alpha)\mathbf{q} \tag{6.7}$$

where $\mathbf{P} = \mathbf{A}\mathbf{D}^{-1}$ is the column normalized transition matrix, $\mathbf{q}$ is the one-hot stochastic vector, and $\mathbf{r}$ is the personalized PageRank vector. The exactness proof of the above solution can be referred to (Yoon et al., 2018b).

Now, suppose we have an original graph whose transition matrix is denoted as $\mathbf{P}$. After inserting some super nodes, we have the new graph whose transition matrix is denoted as $\tilde{\mathbf{P}}$. For dimension consistency, we view the added super nodes as dangling nodes in the original graph, which has no connecting edges.

For the original transition matrix $\mathbf{P}$, we can have the personalized PageRank vector denoted as $\mathbf{r}$. Also, for the new transition matrix $\tilde{\mathbf{P}}$, we can have the personalized PageRank vector denoted as $\tilde{\mathbf{r}}$. Next, given our vector $\mathbf{c} = \alpha \sum_i^\infty \alpha^i \tilde{\mathbf{P}}^i (\tilde{\mathbf{P}} - \mathbf{P})\mathbf{r}$, we need to prove that $\tilde{\mathbf{r}} = \mathbf{c} + \mathbf{r}$. Since vectors $\tilde{\mathbf{r}}$ and $\mathbf{r}$ are non-negative personalized PageRank vector, i.e., $\|\tilde{\mathbf{r}}\|_1 = \|\mathbf{r}\|_1 = 1$, then the sum of vector $\mathbf{c}$ is 0.

Motivated by Theorem 3.2 in Yoon et al. (2018a), we next expand vector $\mathbf{c} = \alpha \sum_i^\infty \alpha^i \tilde{\mathbf{P}}^i (\tilde{\mathbf{P}} - \mathbf{P})\mathbf{r}$:

$$
\begin{aligned}
\mathbf{c} &= \sum_i^\infty \alpha^i \tilde{\mathbf{P}}^i \alpha (\tilde{\mathbf{P}} - \mathbf{P})\mathbf{r} \\
&= \sum_{i=0}^\infty \left( \left( (\alpha\tilde{\mathbf{P}})^{i+1} \sum_{j=0}^\infty (\alpha\mathbf{P})^j (1-\alpha)\mathbf{q} \right) - \left( (\alpha\tilde{\mathbf{P}})^i \sum_{j=0}^\infty (\alpha\mathbf{P})^{j+1} (1-\alpha)\mathbf{q} \right) \right) \\
&= \sum_{i=0}^\infty \left( \left( (\alpha\tilde{\mathbf{P}})^{i+1} \sum_{j=0}^\infty (\alpha\mathbf{P})^j (1-\alpha)\mathbf{q} \right) - \left( (\alpha\tilde{\mathbf{P}})^i \sum_{j=0}^\infty (\alpha\mathbf{P})^j (1-\alpha)\mathbf{q} \right) + \left( (\alpha\tilde{\mathbf{P}})^i (1-\alpha)\mathbf{q} \right) \right) \\
&= \sum_{i=0}^\infty (\alpha\tilde{\mathbf{P}})^{i+1} \sum_{j=0}^\infty (\alpha\mathbf{P})^j (1-\alpha)\mathbf{q} - \sum_{i=0}^\infty (\alpha\tilde{\mathbf{P}})^i \sum_{j=0}^\infty (\alpha\mathbf{P})^j (1-\alpha)\mathbf{q} + \sum_{i=0}^\infty (\alpha\tilde{\mathbf{P}})^i (1-\alpha)\mathbf{q} \\
&= \sum_{i=0}^\infty (\alpha\tilde{\mathbf{P}})^i \sum_{j=0}^\infty (\alpha\mathbf{P})^j (1-\alpha)\mathbf{q} - \sum_{j=0}^\infty (\alpha\mathbf{P})^j (1-\alpha)\mathbf{q} - \sum_{i=0}^\infty (\alpha\tilde{\mathbf{P}})^i \sum_{j=0}^\infty (\alpha\mathbf{P})^j (1-\alpha)\mathbf{q} + \sum_{i=0}^\infty (\alpha\tilde{\mathbf{P}})^i (1-\alpha)\mathbf{q} \\
&= - \sum_{j=0}^\infty (\alpha\mathbf{P})^j (1-\alpha)\mathbf{q} + \sum_{i=0}^\infty (\alpha\tilde{\mathbf{P}})^i (1-\alpha)\mathbf{q}
\end{aligned}
$$

$$(6.8)$$

Then, based on Eq. 6.7, it can be observed that $\mathbf{c} + \mathbf{r} = \tilde{\mathbf{r}}$. Moreover, since $\|\mathbf{P}\| = \|\tilde{\mathbf{P}}\| = 1$, then $\mathbf{c}$ converge to a constant.

**Theorem 6.2** *The efficiency of getting the token list in VCR-Graphormer is $O(m + k\log k)$, where $m$ is the number of edges in the input graph $G$, and $k = max(NNZ(\bar{\mathbf{r}}_u), NNZ(\hat{\mathbf{r}}_u))$, where $NNZ(\cdot)$ denotes the number of non-zero entries in a vector. The complexity of self-attention in VCR-Graphormer is $O((1 + L + \bar{k} + \hat{k})^2)$ based on Eq. 3.3.*

The general insights from Theorem 6.2 state the computational complexity of the proposed graph transformer, which is less than the quadratic attention complexity of most graph transformers.

### 6.2.2 PROOF OF THEOREM 6.2

For getting the token list as stated in Eq. 3.3, computing the second component i.e., $\mathbf{PX}$ costs $O(m)$, where $m$ is the number of edges in the input graph. Then, for computing the third and the fourth components, the analysis is similar. Taking the third component computation as an example, we have below analysis. First, using Metis (Karypis and Kumar, 1998) to partition the input graph into arbitrary clusters costs $O(m)$, suppose the number of nodes $n$ is smaller than the number of edges $m$, and the number of partitions $s$ is way smaller than the number of edges $m$ [4].

Second, approximately solving the personalized PageRank with Push Operations costs constant time complexity (Lemma 2 in (Andersen et al., 2006) and Proposition 2 in (Ohsaka et al., 2015)), which constant is only related to error tolerance $\epsilon$ and dumping factor $\alpha$ but not the size of the input graph. We denote the constant as $c$ in the following. Since Push Operation (Andersen et al., 2006) is a local operation, most entries in the PageRank vector $\mathbf{r}_u$ are zero (or near zero and can be omitted). Denote the number of non-zero entries in $\mathbf{r}_u$ as $k$, then sorting them costs $O(k\log k)$. For each node $u \in \{1, \ldots, n\}$, we totally need $O(n \cdot c \cdot k\log k)$. Since each node's personalized PageRank computation is independent, we can parallelize the computation (Fu and He, 2021). Denote we have $w$ parallel workers, the final complexity is $O(\frac{n}{w} \cdot c \cdot k\log k)$. When number of workers is sufficient, we can have $O(k\log k)$.

### 6.3 HETEROPHILOUS EXPERIMENTS ON LARGE-SCALE DATASETS

The heterophilous arxiv-year data is from the benchmark (Lim et al., 2021), the performance is based on the given split of the benchmark. In addition to the baselines provided by the benchmark like

---

[4] http://glaros.dtc.umn.edu/gkhome/node/419

MixHop (Abu-El-Haija et al., 2019), GCNII (Chen et al., 2020), GPR-GNN (Chien et al., 2021), we include other heterophilous graph deep learning methods WRGAT (Suresh et al., 2021), GGCN (Yan et al., 2022), ACM-GCN (Luan et al., 2021), GloGNN (Li et al., 2022) and GloGNN++ (Li et al., 2022).

Table 6: Node Classification Accuracy on Large-Scale Heterophilous Graph Dataset (Without Minibatching)

|  | arXiv-year |
| --- | --- |
| Edge Homophily | 0.22 |
| #Nodes | 169,343 |
| #Edges | 1,166,243 |
| #Features | 128 |
| #Classes | 5 |
| MLP | $36.70 \pm 0.21$ |
| L Prop 1-hop | $43.42 \pm 0.17$ |
| L Prop 2-hop | $46.07 \pm 0.15$ |
| LINK | $53.97 \pm 0.18$ |
| SGC 1-hop | $32.83 \pm 0.13$ |
| SGC 2-hop | $32.27 \pm 0.06$ |
| C&S 1-hop | $44.51 \pm 0.16$ |
| C&S 2-hop | $49.78 \pm 0.26$ |
| GCN | $46.02 \pm 0.26$ |
| GAT | $46.05 \pm 0.51$ |
| GCNJK | $46.28 \pm 0.29$ |
| GATJK | $45.80 \pm 0.72$ |
| APPNP | $38.15 \pm 0.26$ |
| $H_2$GCN | $49.09 \pm 0.10$ |
| MixHop | $51.81 \pm 0.17$ |
| GPR-GNN | $45.07 \pm 0.21$ |
| GCNII | $47.21 \pm 0.28$ |
| WRGAT | OOM |
| GGCN | OOM |
| ACM-GCN | $47.37 \pm 0.59$ |
| LINKX (Minibatch) | $53.74 \pm 0.27$ |
| GloGNN | $54.68 \pm 0.34$ |
| GloGNN++ | $54.79 \pm 0.25$ |
| VCR-Graphormer | $54.15 \pm 0.09$ |

Table 7: Node Classification Accuracy on Large-Scale Heterophilous Graph Dataset (With Mini-batching)

|  | arXiv-year |
| --- | --- |
| Edge Homophily | 0.22 |
| #Nodes | 169,343 |
| #Edges | 1,166,243 |
| #Features | 128 |
| #Classes | 5 |
| MLP Minibatch | $36.89 \pm 0.11$ |
| LINK Minibatch | $53.76 \pm 0.28$ |
| GCNJK-Cluster | $44.05 \pm 0.11$ |
| GCNJK-SAINT-Node | $44.30 \pm 0.22$ |
| GCNJK-SAINT-RW | $47.40 \pm 0.17$ |
| MixHop-Cluster | $48.41 \pm 0.31$ |
| MixHop-SAINT-Node | $44.84 \pm 0.18$ |
| MixHop-SAINT-RW | $50.55 \pm 0.20$ |
| LINKX Minibatch | $53.74 \pm 0.27$ |
| VCR-Graphormer | $54.15 \pm 0.09$ |

## 6.4 EFFICIENCY COMPARISON

On the largest dataset Amazon2M, we did the efficiency comparison. The eigendesompostion is based on DGL library. DGL itself is a Python library, but its underlying computations may involve languages like C and C++, and high-performance linear algebra libraries like SciPy and Fortran to ensure efficient numerical computations. Even without strong support, our PageRank sampling realized by Python (with parallel computing as theoretically designed in the paper) is slightly faster. Moreover, the reason why the content-based virtual node sampling is faster (i.e., $\sim$620 v.s. $\sim$409) is that the number of content-based virtual super nodes depends on the number of labels, which is usually less than the number of structure-based virtual super nodes. Hence, its PPR convergence is faster. The below time consumption contains the METIS partitioning time realized by DGL.

Table 8: Time Consumption on Amazon2M Dataset

| Method | Time (s) |
| --- | --- |
| Eigendecomposition (by DGL library) | $\sim$682 |
| Virtual Structure Node based Parallel PageRank Sampling for Each Node (by Python) | $\sim$620 |
| Virtual Content Node based Parallel PageRank Sampling for Each Node (by Python) | $\sim$409 |

## 6.5 IMPLEMENTATION DETAILS

For the second component in Eq. 3.3, we used the symmetrically normalized transition matrix $\mathbf{P} = \mathbf{D}^{-\frac{1}{2}}\mathbf{A}\mathbf{D}^{-\frac{1}{2}}$. For the thrid and fourth component in Eq. 3.3, we used row normalized transition matrix $\mathbf{P} = \mathbf{D}^{-1}\mathbf{A}$. Also, for connecting content-based super nodes, we only use the label of the training set. The experiments are performed on a Linux machine with a single NVIDIA Tesla V100 32GB GPU.

Reddit dataset is a previously existing dataset extracted and obtained by a third party that contains preprocessed comments posted on the social network Reddit and hosted by Pushshift.io.

