# OpenReview forum: "VCR-Graphormer: A Mini-batch Graph Transformer via Virtual Connections"
_ICLR.cc/2024/Conference — ICLR 2024 poster_

### Official Review · Reviewer_QU1U · 2023-10-25

**Soundness:** 2 fair
**Presentation:** 3 good
**Contribution:** 2 fair
**Rating:** 5
**Confidence:** 5

**Summary:**

This work try to combine PPR and transformer together to perform mini-batch training.

**Strengths:**

1. The paper aims to solve the mini-batch training of graph transformer.

2. the paper is well written and easy to follow.

3. The performance of this work is good, comparing to the baseline.

**Weaknesses:**

1. The novelty of this work is not high, as it mainly combines random walk with attention. Tokenize the graph to sequence use random walk is common used.

2. The code is not provided, its reproducibility is unclear.

**Questions:**

1. The novelty of this work is not high, as it mainly combines random walk with attention. Tokenize the graph to sequence use random walk is common used.

2. The code is not provided, its reproducibility is unclear.

---

> ### Author Response · Authors · 2023-11-23
> **Addressing the concerns of Reviewer QU1U**
>
> Thanks very much for your raised concerns. They are addressed as follows.
>
> * First, the code will be released upon the publication of the paper.
>
> - Second, we would like to mention that we (I) first analyzed the standard for graph tokenization with the theory derivation (i.e., Thm 3.1 and Table 1) and then (II) invented a virtual node based pagerank for graph tokenization with theoretical and empirical evaluation, in order to reduce the quadratic complexity of graph transformer for node classification in different settings like homophilous and heterophilous graphs.
>
> * Finally, during this rebuttal period, our paper gets improved by including more experiments with more SOTA algorithms, larger heterophilous datasets, and efficiency experiments. All the updates are included in the revised paper and marked blue.

---

### Official Review · Reviewer_5Wvh · 2023-10-31

**Soundness:** 2 fair
**Presentation:** 3 good
**Contribution:** 3 good
**Rating:** 6
**Confidence:** 5

**Summary:**

To scale the Transformer to large graphs, this paper proposes a new graph Transformer, called VCR-Graphormer. The key idea of the proposed method is to sample a token list constructed by related nodes on graphs for each target node. In this way, the mini-batch training strategy could be adopted to reduce the training cost. Moreover, the authors leverage techniques like PPR and virtual connections to preserve both local, global, long-range, and heterophilous information.

**Strengths:**

1.	This paper summarizes four metrics for graph tokenization methods.
2.	This paper leverages existing techniques, like PPR and the graph partition method, to generate the token list for each target node.
3.	This paper provides several theoretical analyses for the proposed method.
4.	Empirical results on different scale datasets seem to indicate the promising performance of the proposed method.

**Weaknesses:**

1.	Several recent studies on designing graph Transformers with node sampling or node clustering are ignored.
2.	Experimental results are inefficient in demonstrating the merits of the proposed method.

**Questions:**

1.	I think the proposed method belongs to the line of designing scalable graph Transformers via node sampling. Hence, several necessary studies [1,2,3] on this research topic should be cited and discussed in the paper. [1] and [3] leverage various node sampling strategies to obtain the token list for each node, where a super node-based strategy is also adopted in [3] to preserve the global information. [2] leverage the graph partition-based strategy to reduce the training cost of the Transformer model. These researches are highly related to the proposed method, especially [1] and [3].
2.	Based on Q1, I think it is necessary to compare the performance of the proposed method with [1] and [3] to demonstrate the superiority of the proposed method for constructing the node sequence.
3.	I think the results on heterophilous graph datasets are inefficient in demonstrating that the proposed method can handle graphs with heterophily. Important baselines [4,5,6] and datasets [7, 8] are not considered in the experiment. In addition, [8] has revealed the drawbacks of the Squirrel dataset. Hence, experiments on heterophilous graph datasets need to be reorganized to support the claim of handling heterophily property.
4.	Similarly, the authors have highlighted that efficiency is one of the important metrics for graph tokenization methods. So, the necessary experiment is required to support the above claim.
5. Does the sampling node set of the third component contain super nodes? If the answer is yes, how do you initialize the features of super nodes?



[1] Zhao et al. Gophormer: Ego-Graph Transformer for Node Classification. arXiv 2021.

[2] Kuang et al. Coarformer: Transformer for large graph via graph coarsening. arXiv 2021.

[3] Zhang et al. Hierarchical graph transformer with adaptive node sampling. NeurIPS 2022.

[4] Bo et al. Beyond Low-frequency Information in Graph Convolutional Networks. AAAI 2021.

[5] Chien et al. Adaptive universal generalized pagerank graph neural network. ICLR 2022.

[6] Li et al. Finding Global Homophily in Graph Neural Networks When Meeting Heterophily. ICML 2022.

[7] Lim et al. Large Scale Learning on Non-Homophilous Graphs: New Benchmarks and Strong Simple Methods. NeurIPS 2021.

[8] Platonov et al. A critical look at the evaluation of GNNs under heterophily: are we really making progress? ICLR 2023.

---

> ### Author Response · Authors · 2023-11-23
> **Addressing the concerns of Reviewer 5Wvh (Part I)**
>
> Thanks very much for your review. Your raised concerns are addressed as follows.
>
> * First, all the suggested literature is added and discussed in the revised paper.
>
> - Second, we add the benchmark large-scale heterophilous datasets [7] and include a comprehensive comparison with more baselines. The table is shown below. In short, our proposed method can still outperform the competitors. More details are included in the revised paper and colored blue in Appendix 6.4.
>
> Table 1. Node Classification Accuracy on Large-Scale Heterophilous Graph Dataset (Without Minibatching)
> |                               | arXiv-year                 |
> |-------------------------------|---------------------------|
> | **Edge Homophily**            | 0.22                      |
> | **#Nodes**                    | 169,343                   |
> | **#Edges**                    | 1,166,243                 |
> | **#Features**                 | 128                       |
> | **#Classes**                  | 5                         |
> | **MLP**                       | 36.70 $\pm$ 0.21         |
> | **L Prop 1-hop**              | 43.42 $\pm$ 0.17         |
> | **L Prop 2-hop**              | 46.07 $\pm$ 0.15         |
> | **LINK**                      | 53.97 $\pm$ 0.18         |
> | **SGC 1-hop**                 | 32.83 $\pm$ 0.13         |
> | **SGC 2-hop**                 | 32.27 $\pm$ 0.06         |
> | **C\&S 1-hop**                | 44.51 $\pm$ 0.16         |
> | **C\&S 2-hop**                | 49.78 $\pm$ 0.26         |
> | **GCN**                       | 46.02 $\pm$ 0.26         |
> | **GAT**                       | 46.05 $\pm$ 0.51         |
> | **GCNJK**                     | 46.28 $\pm$ 0.29         |
> | **GATJK**                     | 45.80 $\pm$ 0.72         |
> | **APPNP**                     | 38.15 $\pm$ 0.26         |
> | **H$_{2}$GCN**                | 49.09 $\pm$ 0.10         |
> | **MixHop**                    | 51.81 $\pm$ 0.17         |
> | **GPR-GNN**                   | 45.07 $\pm$ 0.21         |
> | **GCNII**                     | 47.21 $\pm$ 0.28         |
> | **WRGAT**                     | OOM                       |
> | **GGCN**                      | OOM                       |
> | **ACM-GCN**                   | 47.37 $\pm$ 0.59         |
> | **LINKX (Minibatch)**         | 53.74 $\pm$ 0.27         |
> | **GloGNN**                    | 54.68 $\pm$ 0.34         |
> | **GloGNN++**                  | 54.79 $\pm$ 0.25         |
> | **VCR-Graphormer**            | 54.15 $\pm$ 0.09         |
>
> Table 2. Node Classification Accuracy on Large-Scale Heterophilous Graph Dataset (With Minibatching)
> |                               | arXiv-year                |
> |-------------------------------|---------------------------|
> | **Edge Homophily**            | 0.22                      |
> | **#Nodes**                    | 169,343                   |
> | **#Edges**                    | 1,166,243                 |
> | **#Features**                 | 128                       |
> | **#Classes**                  | 5                         |
> | **MLP Minibatch**             | 36.89 $\pm$ 0.11         |
> | **LINK Minibatch**            | 53.76 $\pm$ 0.28         |
> | **GCNJK-Cluster**             | 44.05 $\pm$ 0.11         |
> | **GCNJK-SAINT-Node**          | 44.30 $\pm$ 0.22         |
> | **GCNJK-SAINT-RW**            | 47.40 $\pm$ 0.17         |
> | **MixHop-Cluster**            | 48.41 $\pm$ 0.31         |
> | **MixHop-SAINT-Node**         | 44.84 $\pm$ 0.18         |
> | **MixHop-SAINT-RW**           | 50.55 $\pm$ 0.20         |
> | **LINKX Minibatch**           | 53.74 $\pm$ 0.27         |
> | **VCR-Graphormer**            | 54.15 $\pm$ 0.09         |

---

> > ### Author Response · Authors · 2023-11-23
> > **Addressing the concerns of Reviewer 5Wvh (Part II)**
> >
> > * Third, in addition to the theoretical computational complexity analysis, the empirical efficiency comparison is conducted and updated. This running time question is also shared by Reviewer jsyZ. Therefore, on the largest dataset Amazon2M, we did the efficiency comparison. The eigendecompostion is based on DGL library. DGL itself is a Python library, but its underlying computations may involve languages like C and C++, and high-performance linear algebra libraries like SciPy and Fortran to ensure efficient numerical computations. Even without strong support, our PageRank sampling realized by Python (with parallel computing as theoretically designed in the paper) is slightly faster. Moreover, the reason why the content-based virtual node sampling is faster (i.e., ~620 v.s. ~409) is that the number of content-based virtual super nodes depends on the number of labels, which is usually less than the number of structure-based virtual super nodes. Hence, its PPR convergence is faster.
> >
> > Table 3. Time Consumption on Amazon2M Dataset (added in Appendix 6.5)
> > | Method | Time (s) |
> > |--------|----------|
> > | **Eigendecomposition (by DGL library)** | $\sim$682 |
> > | **Virtual Structure Node based Parallel PageRank Sampling for Each Node (by Python)** | $\sim$620 |
> > | **Virtual Content Node based Parallel PageRank Sampling for Each Node (by Python)** | $\sim$409 |
> >
> > * Fourth, no matter structure-based virtual super nodes or content-based virtual super nodes, they do not participate in the sampling set. Therefore, their features are initialized as empty vectors and not updated. Their role is to decrease the pairwise distance to further extend the narrow receptive field.

---

> > > ### Comment · Reviewer_5Wvh · 2023-11-23
> > >
> > > Thanks for the response. My opinion of the paper remains unchanged and I will keep my score.

---

### Official Review · Reviewer_82Qp · 2023-11-01

**Soundness:** 3 good
**Presentation:** 3 good
**Contribution:** 3 good
**Rating:** 6
**Confidence:** 4

**Summary:**

In this work the authors propose a way for effective mini-batch training of graph transformers. In their approach, namely Virtual Connection Ranking Graph Transformer (VCR-Graphformer), they build a special token list for each input target node as input to a transformer. This list consists of four components including:
- (1) the node features of the target node,
- (2) propagated features for a number of random walk steps and features of nodes with top Personalized PageRank (PPR) when PPR is run on extended graphs (rewiring) with extra virtual nodes connected to original nodes either
- (3) in the same cluster (structure-based global information) or
- (4) carrying the same label (content-based global information).

They provide theoretical justification for their direction, experiment with their mini-batching approach over a collection of small, large and heterophilous graph datasets, employing various Graph Neural Network (GNN) and Graph Transformer (GT) architectures as baselines, and report on (a) superior competitive node classification accuracies for their VCR-Graphormer and (b) ablation studies clarifying the role of components in the token list and parameter choices.

**Strengths:**

- The presentation is easy to follow and the intuition of the approach well supported.

- Types, number of datasets and baseline models are adequate for demonstrating the efficacy of the approach for the node classification task in particular. Ablation studies are very informative (Figures 3 and 4).

**Weaknesses:**

- The inclusion of additional graph learning tasks (edge/graph classification) would further establish the validity/generality of the token list preparation approach proposed.

**Questions:**

- VCR-Graphormer seems to be an input preprocessing technique (preparation of the token list for input to standard transfromer layers as in Eq (3.4)) that is compatible with the target training mode (mini-batching) rather than a graph transformer architecture (which is what the reader would possibly expect when coming across the term "VCR-Transformer"). Perhaps emphasizing this view early in the presentation would be beneficial in comprehending the overall idea?

- Do you identify any obstacles for this technique being effective for other (large-scale) graph learning tasks (not node classification)?

- VCR-Graphormer supports effective heterophilous graph node label learning through one of its token list components. Would this also promote homophilous graph learning?

Minor/typos
- Page 3: share the exactly same -> share exactly the same
- Page 7: eigendecompostion -> eigendecomposition
- Page 9: strcuture -> structure
- Page 9: Targeting node-level tasks like graph classification -> ? (graph classification is not a node-level task)

---

> ### Author Response · Authors · 2023-11-23
> **Addressing the concerns of Reviewer 82Qp**
>
> Thanks so much for your actionable questions. They are addressed below.
>
> * First, our proposed method can be viewed as the preprocessing technique designed to prepare the token list for input to standard transformer layers, which aims to save the quadratic complexity of the graph transformer's dense attention. We will emphasize this more at the beginning of the paper.
>
> - Second, so far, we have the experiments for node classification but in different settings like homophilous and heterophilous datasets. Table 4 and Table 5 are the experiments on the homophilous datasets. Figure 2 and the newly added Table 6&7 are for heterophilous datasets. Extending to more general applications like link prediction and graph classification with graph transformers is a very interesting direction, and we will explore it in the future.
>
> * Third, your pointed-out minor typos are corrected in the updated paper.
>
> - Finally, during this rebuttal period, our paper gets improved by including more experiments with more SOTA algorithms, larger heterophilous datasets, and efficiency experiments.

---

### Official Review · Reviewer_jsyZ · 2023-11-01

**Soundness:** 2 fair
**Presentation:** 1 poor
**Contribution:** 3 good
**Rating:** 5
**Confidence:** 4

**Summary:**

The paper proposes VCR-Graphormer, which resolves the computational challenges of graph transformers in large-scale graphs. The authors tokenize nodes using personalized PageRank (PPR), enabling mini-batch training. Additionally, they introduce virtual connections in the graph to encode local and global contexts, long-range interactions, and heterophilous signals. This approach reduces computational complexity and outperforms or is on par with existing methods in node classification across 12 datasets.

**Strengths:**

The scalable graph transformer is a hot, interesting, and important field in our research community. Training with mini-batches is memory-efficient by nature. The experiments are extensive (but focus on homophilous datasets). The proposed method outperforms baselines by a large margin, especially for heterophilous graphs.

**Weaknesses:**

- The paper is hard to follow. Abuse of notations in Eq 3.1, 3.2, and 3.3 might be confusing for readers. Please formally define the operations or use well-known operations. Is {.} a set or a list? Please proper set-builder notations. Please put l (l-th step) to the name of the variable, r_u? Is the concatenation operator applied to both scalars and vectors? The cardinality of T_u in Eq 3.3 is 4 when you note it like this. Plus, it would be nice if the authors explained what insights we can see in a series of theorems 3.2 and 3.3.
- The eigendecomposition is not a core component in NAGphormer. NAGphormer without structural encoding has shown no bad results. We can choose other cheap structural encodings rather than eigendecomposition, for example, PPR vectors as the authors say. Thus, the superiority of the proposed method against NAGphormer (e.g., cubic complexity) should be rewritten focusing on the core components (Hop2Token).
- The performance increase in homophilous and large-scale datasets is marginal. VCR-Graformer's additional modules do not seem to be effective for these datasets. Instead, the proposed method is effective for heterophilous datasets. However, these datasets are small-scale thus, only evaluating small parts of this model (that targets large-scale graphs). It would be nice if the authors can use large-scale heterophilous datasets [Lim, Derek, et al.]. In addition, experiments to evaluate modeling long-range dependency (the third component) are not conducted.
- Runtime evaluation on efficiency is required. Although the time complexity is decreased in theory, the actual computations can be increased by METIS and PPR on three types of graphs.

## References
- [Lim, Derek, et al.] Lim, Derek, et al. "Large scale learning on non-homophilous graphs: New benchmarks and strong simple methods." Advances in Neural Information Processing Systems 34 (2021): 20887-20902.

**Questions:**

N/A

---

> ### Author Response · Authors · 2023-11-23
> **Addressing the concerns of Reviewer jsyZ (Part I)**
>
> Thanks for your actionable suggestions. They are addressed as follows.
>
> * First, the suggestions regarding some symbols and notation are addressed in the updated paper and marked blue. Moreover, the general insights from Thm 3.2 show the probability mass transfer from faded ordinary nodes to brightened ordinary nodes, which further illustrates that previous hardly-reachable nodes are obtaining more weight to extend the narrow receptive field. The general insights from Thm 3.3 state the computational complexity of the proposed graph transformer, which is less than the quadratic attention complexity of most graph transformers.
>
> - Second, thanks for the reminder of the role of eigendecomposition in NAGphormer. We have updated Table 1 in the paper and made the necessary explanation in the context marked blue.
>
> * Third, according to the suggestion, we test the performance on large-scale datasets [Lim et al, 2021]. The experiments are shown below. In short, our proposed method can still outperform the competitors. More details are included in the paper and marked blue in Appendix 6.4.
>
> Table 1. Node Classification Accuracy on Large-Scale Heterophilous Graph Dataset (Without Minibatching)
> |                               | arXiv-year                 |
> |-------------------------------|---------------------------|
> | **Edge Homophily**            | 0.22                      |
> | **#Nodes**                    | 169,343                   |
> | **#Edges**                    | 1,166,243                 |
> | **#Features**                 | 128                       |
> | **#Classes**                  | 5                         |
> | **MLP**                       | 36.70 $\pm$ 0.21         |
> | **L Prop 1-hop**              | 43.42 $\pm$ 0.17         |
> | **L Prop 2-hop**              | 46.07 $\pm$ 0.15         |
> | **LINK**                      | 53.97 $\pm$ 0.18         |
> | **SGC 1-hop**                 | 32.83 $\pm$ 0.13         |
> | **SGC 2-hop**                 | 32.27 $\pm$ 0.06         |
> | **C\&S 1-hop**                | 44.51 $\pm$ 0.16         |
> | **C\&S 2-hop**                | 49.78 $\pm$ 0.26         |
> | **GCN**                       | 46.02 $\pm$ 0.26         |
> | **GAT**                       | 46.05 $\pm$ 0.51         |
> | **GCNJK**                     | 46.28 $\pm$ 0.29         |
> | **GATJK**                     | 45.80 $\pm$ 0.72         |
> | **APPNP**                     | 38.15 $\pm$ 0.26         |
> | **H$_{2}$GCN**                | 49.09 $\pm$ 0.10         |
> | **MixHop**                    | 51.81 $\pm$ 0.17         |
> | **GPR-GNN**                   | 45.07 $\pm$ 0.21         |
> | **GCNII**                     | 47.21 $\pm$ 0.28         |
> | **WRGAT**                     | OOM                       |
> | **GGCN**                      | OOM                       |
> | **ACM-GCN**                   | 47.37 $\pm$ 0.59         |
> | **LINKX (Minibatch)**         | 53.74 $\pm$ 0.27         |
> | **GloGNN**                    | 54.68 $\pm$ 0.34         |
> | **GloGNN++**                  | 54.79 $\pm$ 0.25         |
> | **VCR-Graphormer**            | 54.15 $\pm$ 0.09         |
>
> Table 2. Node Classification Accuracy on Large-Scale Heterophilous Graph Dataset (With Minibatching)
> |                               | arXiv-year                |
> |-------------------------------|---------------------------|
> | **Edge Homophily**            | 0.22                      |
> | **#Nodes**                    | 169,343                   |
> | **#Edges**                    | 1,166,243                 |
> | **#Features**                 | 128                       |
> | **#Classes**                  | 5                         |
> | **MLP Minibatch**             | 36.89 $\pm$ 0.11         |
> | **LINK Minibatch**            | 53.76 $\pm$ 0.28         |
> | **GCNJK-Cluster**             | 44.05 $\pm$ 0.11         |
> | **GCNJK-SAINT-Node**          | 44.30 $\pm$ 0.22         |
> | **GCNJK-SAINT-RW**            | 47.40 $\pm$ 0.17         |
> | **MixHop-Cluster**            | 48.41 $\pm$ 0.31         |
> | **MixHop-SAINT-Node**         | 44.84 $\pm$ 0.18         |
> | **MixHop-SAINT-RW**           | 50.55 $\pm$ 0.20         |
> | **LINKX Minibatch**           | 53.74 $\pm$ 0.27         |
> | **VCR-Graphormer**            | 54.15 $\pm$ 0.09         |

---

> ### Author Response · Authors · 2023-11-23
> **Addressing the concerns of Reviewer jsyZ (Part II)**
>
> * Fourth, the efficiency comparison is conducted and updated. On the largest dataset Amazon2M, we did the efficiency comparison. The eigendecompostion is based on DGL library. DGL itself is a Python library, but its underlying computations may involve languages like C and C++, and high-performance linear algebra libraries like SciPy and Fortran to ensure efficient numerical computations. Even without strong support, our PageRank sampling realized by Python (with parallel computing as theoretically designed in the paper) is slightly faster. Moreover, the reason why the content-based virtual node sampling is faster (i.e., ~620 v.s. ~409) is that the number of content-based virtual super nodes depends on the number of labels, which is usually less than the number of structure-based virtual super nodes. Hence, its PPR convergence is faster. The below running time comparison contains the METIS partitioning time realized by DGL.
>
> Table 3. Time Consumption on Amazon2M Dataset (added in Appendix 6.5)
>
> | Method | Time (s) |
> |--------|----------|
> | **Eigendecomposition (by DGL library)** | $\sim$682 |
> | **Virtual Structure Node based Parallel PageRank Sampling for Each Node (by Python)** | $\sim$620 |
> | **Virtual Content Node based Parallel PageRank Sampling for Each Node (by Python)** | $\sim$409 |

---

> ### Comment · Reviewer_jsyZ · 2023-11-23
>
> > First, the suggestions regarding some symbols and notation are addressed in the updated paper and marked blue.
>
> In my opinion, notations are still confusing: Eq 3.1, 3.2, 3.3, and r_u. The operator still needs to be better defined. Using conventional notations can improve readability for future readers.
>
> > Moreover, the general insights from Thm 3.2 show the probability mass transfer from faded ordinary nodes to brightened ordinary nodes, which further illustrates that previous hardly-reachable nodes are obtaining more weight to extend the narrow receptive field. The general insights from Thm 3.3 state the computational complexity of the proposed graph transformer, which is less than the quadratic attention complexity of most graph transformers.
>
> Could you include these explanations in the next revision?
>
> > Table 3. Time Consumption on Amazon2M Dataset (added in Appendix 6.5)
>
> 1. To be clear, if we use both PageRank sampling, is it correct that it takes longer than eigendecomposition?
> 2. In practice, eigendecomposition is computed once before training and cached for future inferences. We do not need to re-compute it for transductive node classification tasks. How about PageRank sampling? Do we have to run multiple times for PageRank sampling?

---

### Meta-Review · Area_Chair_1QpM · 2023-12-07

**Metareview:**

This work proposes VCR-Graphormer, a procedure to scale the Transformer architecture to larger graphs. The proposed method introduces a new approach for sampling token lists from related graph nodes, tailored to each target node. This strategy simplifies the implementation of mini-batch training, significantly reducing training costs. Additionally, the method incorporates techniques such as Personalized PageRank and virtual connections, effectively capturing and preserving local, global, long-range, and heterophilous information within the graph structure.

* The methodological contribution is rather small. The main contribution is a decent engineering effort and a clear description of the reasoning driving the decisions.
* The proposed method does not seem clearly superior to NAGphormer experimentally, rather, it is an alternative. By avoiding a spectral decomposition the model can be seen as more scalable (but spectral decompositions of the top k components can also be performed by RandNLA algorithms which are fast and scalable [not $O(n^3)]$ as the authors describe in their work]).

**Justification For Why Not Higher Score:**

This is a borderline paper that could be accepted, it is not a spotlight paper.

**Justification For Why Not Lower Score:**

The methodological contribution is rather small. The main contribution is a decent engineering effort and a clear description of the reasoning driving the decisions. It is a decent contribution from an engineering standpoint towards better graph transformer models.

---

### Decision · Program_Chairs · 2024-01-16

Accept (poster)